# GENERATIVE ADVERSARIAL SELF-IMITATION LEARNING

## ABSTRACT

This paper explores a simple regularizer for reinforcement learning by proposing *Generative Adversarial Self-Imitation Learning* (GASIL), which encourages the agent to imitate past good trajectories via generative adversarial imitation learning framework. Instead of directly maximizing rewards, GASIL focuses on reproducing past good trajectories, which can potentially make long-term credit assignment easier when rewards are delayed. GASIL can be easily combined with any policy gradient objective by using GASIL as a learned reward shaping function. Our experimental results show that GASIL improves the performance of proximal policy optimization on 2D Point Mass and MuJoCo environments with delayed reward and stochastic dynamics.

## 1 INTRODUCTION

A major component of Reinforcement learning (RL) is the temporal credit assignment problem that amounts to figuring out which action in a state leads to a better outcome in the future. Different RL algorithms have different forms of objectives to solve this problem. For example, policy gradient approaches learn to directly adapt the policy to optimize the RL objective (i.e., maximizing cumulative rewards), while value-based approaches (e.g., Q-Learning (Watkins & Dayan, 1992)) estimate long-term future rewards and induce a policy from it. Policies optimized for different objectives many have different learning dynamics, which end up with different sub-optimal policies in complex environments, though all of these objectives are designed to maximize cumulative rewards.

In this paper, we explore a simple regularizer for RL, called *Generative Adversarial Self-Imitation Learning* (GASIL). Instead of directly maximizing rewards, GASIL aims to imitate past good trajectories that the agent has generated using generative adversarial imitation learning framework (Ho & Ermon, 2016). GASIL solves the temporal credit assignment problem by learning a discriminator which discriminates between the agent's current trajectories and good trajectories in the past, while the policy is trained to make it hard for the discriminator to distinguish between the two types of trajectories by imitating good trajectories. GASIL can potentially make long-term temporal credit assignment easier when reward signal is delayed, because reproducing certain trajectories is often much easier than maximizing long-term delayed rewards. GASIL can be interpreted as an optimal reward learning algorithm (Singh et al., 2009; Sorg et al., 2010), where the discriminator acts as a learned reward function which provides dense rewards for the agent to reproduce relatively better trajectories. Thus, it can be used as a shaped reward function and combined with any RL algorithms.

Our empirical results on 2D Point Mass and OpenAI Gym MuJoCo tasks (Brockman et al., 2016; Todorov et al., 2012) show that GASIL improves the performance of proximal policy optimization (PPO) (Schulman et al., 2017), especially when rewards are delayed. We also show that GASIL is robust to stochastic dynamics to some extent in practice.

## 2 RELATED WORK

**Generative adversarial learning** Generative adversarial networks (GANs) (Goodfellow et al., 2014) have been increasingly popular for generative modeling. In GANs, a discriminator is trained to discriminate whether a given sample is drawn from data distribution or model distribution. A generator (i.e., model) is trained to "fool" the discriminator by generating samples that are close to the real data. This adversarial play allows the model distribution to match to the data distribution. This approach has been very successful for image generation and manipulation (Radford et al., 2015;

Reed et al., 2016; Zhu et al., 2017). Recently, Ho & Ermon (2016) proposed generative adversarial imitation learning (GAIL) which extends this idea to imitation learning. In GAIL, a discriminator is trained to discrimiate between optimal trajectories (or expert trajectories) and policy trajectories, while the policy learns to fool the discriminator by imitating optimal trajectories. Our work further extends this idea to RL. Unlike GAN or GAIL setting, however, optimal trajectories are not available to the agent in RL. Instead, our GASIL treats "relatively better trajectories" that the policy has generated as optimal trajectories that the agent should imitate.

**Reward learning** Singh et al. (2009) discussed a problem of learning an internal reward function that is useful across a distribution of environments in an evolutionary context. Simiarly, Sorg et al. (2010) introduced *optimal reward problem* under the motivation that the true reward function defined in the environment may not be optimal for learning, and there exists an optimal reward function that allows learning the desired behavior much quickly. This claim is consistent with the idea of reward shaping (Ng et al., 1999) which helps learning without changing the optimal policy. There has been a few attempts to learn such an internal reward function without domain-specific knowledge in deep RL context (Sorg et al., 2010; Guo et al., 2016; Zheng et al., 2018). Our work is closely related to this line of work in that GASIL learns a discriminator which acts as an interal reward function that allows the agent to learn to maximize external rewards more easily.

**Self-imitation** There has been a line of work that introduces a notion of learning and inducing a good policy by focusing on good experiences that the agent has generated. For example, episodic control (Lengyel & Dayan, 2008; Blundell et al., 2016; Pritzel et al., 2017) and the nearest neighbor policy (Mansimov & Cho, 2017) construct a non-parametric policy directly from the past experience by retreiving similar states in the past and following the best decision made in the past. Instead, our work aims to learn a parametric policy from past good experiences. Self-imitation has been shown to be useful for program synthesis (Liang et al., 2016; Abolafia et al., 2018), where the agent is trained to generate K-best programs generated by itself. Our work proposes a different objective based on generative adversarial learning framework and evaluates it on RL benchmarks. More recently, Goyal et al. (2018) proposed to learn a generative model of preceding states of high-value states (i.e., top-K trajectories) and update a policy to follow the generated trajectories. In contrast, our GASIL directly learns to imitate past good trajectories without learning a generative model. GASIL can be viewed as a generative adversarial extension of *self-imitation learning* (Oh et al., 2018) which updates the policy and the value function towards past better trajectories. Contemporaneously with our work, Gangwani et al. (2018) also proposed the same method as our GASIL, which was independently developed. Most of the previous works listed above including ours may not guarantee policy improvement under certain types of stochastic environments due to its bias towards positive outcome, though they have been shown to work well on existing benchmarks when used as a regularizer. Looking forward, dealing with stochasticity with this type of approach with stronger theoretical guarantees would be an interesting future direction.

## 3 BACKGROUND

Throughout the paper, we consider a finite state space $\mathcal{S}$ and a finite action space $\mathcal{A}$. The goal of RL is to find a policy $\pi \in \Pi : \mathcal{S} \times \mathcal{A} \to [0, 1]$ which maximizes the discounted sum of rewards: $\eta(\pi) = \mathbb{E}_\pi \left[ \sum_{t=0}^{\infty} \gamma^t r_t \right]$ where $\gamma$ is a discount factor and $r_t$ is a reward at time-step $t$.

Alternatively, we can re-write the RL objective $\eta(\pi)$ in terms of occupancy measure. Occupancy measure $\rho_\pi \in \mathcal{D} : \mathcal{S} \times \mathcal{A} \to \mathbb{R}$ is defined as $\rho_\pi(s, a) = \pi(a|s) \sum_{t=0}^{\infty} \gamma^t P(s_t = s|\pi)$. Intuitively, it is a joint distribution of states and actions visited by the agent following the policy $\pi$. It is shown that there is a one-to-one correspondence between the set of policies ($\Pi$) and the set of valid occupancy measures ($\mathcal{D} = \{\rho_\pi : \pi \in \Pi\}$) (Syed et al., 2008). This allows us to write the RL objective in terms of occupancy measure as follows:

$$\eta(\pi) = \mathbb{E}_\pi \left[ \sum_{t=0}^{\infty} \gamma^t r_t \right] = \sum_{s,a} \rho_\pi(s, a) r(s, a). \tag{1}$$

where $r(s, a)$ is the reward for choosing action $a$ in state $s$. Thus, policy optimization amounts to finding an optimal occupancy measure which maximizes rewards due to the one-to-one correspondence between them.

## 3.1 POLICY GRADIENT

Policy gradient methods compute the gradient of the RL objective $\eta(\pi_\theta) = \mathbb{E}_{\pi_\theta} \left[ \sum_{t=0}^{\infty} \gamma^t r_t \right]$. Since $\eta(\pi_\theta)$ is non-differentiable with respect to the parameter $\theta$ when the dynamics of the environment are unknown, policy gradient methods rely on the score function estimator to get an unbiased gradient estimator of $\eta(\pi_\theta)$. A typical form of policy gradient objective is given by:

$$J_{\text{PG}}(\theta) = \mathbb{E}_{\pi_\theta} \left[ \log \pi_\theta(a_t|s_t)\hat{A}_t \right] \tag{2}$$

where $\pi_\theta$ is a policy parameterized by $\theta$, and $\hat{A}_t$ is an advantage estimation at time $t$. Intuitively, the policy gradient objective either increases the probability of the action when the return is higher than expected ($\hat{A}_t > 0$) or decreases the probability when the return is lower than expected ($\hat{A}_t < 0$).

## 3.2 GENERATIVE ADVERSARIAL IMITATION LEARNING

Generative adversarial imitation learning (GAIL) (Ho & Ermon, 2016) is an imitation learning algorithm which aims to learn a policy that can imitate expert trajectories using the idea from generative adversarial network (GAN) (Goodfellow et al., 2014). More specifically, the objective of GAIL for maximum entropy IRL (Ziebart et al., 2008) is defined as:

$$\underset{\theta}{\text{argmin}} \, \underset{\phi}{\text{argmax}} \, \mathcal{L}_{\text{GAIL}}(\theta, \phi) = \mathbb{E}_{\pi_\theta} \left[ \log D_\phi(s, a) \right] + \mathbb{E}_{\pi_E} \left[ \log(1 - D_\phi(s, a)) \right] - \lambda \mathcal{H}(\pi_\theta) \tag{3}$$

where $\pi_\theta, \pi_E$ are a policy parameterized by $\theta$ and an expert policy respectively. $D_\phi(s, a) : \mathcal{S} \times \mathcal{A} \rightarrow [0, 1]$ is a discriminator parameterized by $\phi$. $\mathcal{H}(\pi) = \mathbb{E}_\pi \left[ -\log \pi(a|s) \right]$ is the entropy of the policy. Similar to GANs, the discriminator and the policy play an adversarial game by either maximizing or minimizing the objective $\mathcal{L}_{\text{GAIL}}$, and the gradient of each component is given by:

$$\nabla_\phi \mathcal{L}_{\text{GAIL}} = \mathbb{E}_{\tau_\pi} \left[ \nabla_\phi \log D_\phi(s, a) \right] + \mathbb{E}_{\tau_E} \left[ \nabla_\phi \log(1 - D_\phi(s, a)) \right] \tag{4}$$

$$\nabla_\theta \mathcal{L}_{\text{GAIL}} = \mathbb{E}_{\tau_\pi} \left[ \nabla_\theta \log D_\phi(s, a) \right] - \lambda \mathcal{H}(\pi_\theta) \tag{5}$$

$$= \mathbb{E}_{\tau_\pi} \left[ \nabla_\theta \log \pi_\theta(a|s) Q(s, a) \right] - \lambda \mathcal{H}(\pi_\theta), \tag{6}$$

where $Q(\bar{s}, \bar{a}) = \mathbb{E}_{\tau_\pi} \left[ \log D_\phi(s, a) | s_0 = \bar{s}, a_0 = \bar{a} \right]$, and $\tau_\pi, \tau_E$ are trajectories sampled from $\pi_\theta$ and $\pi_E$ respectively. Intuitively, the discriminator $D_\phi$ is trained to discriminate between the policy's trajectories ($\tau_\pi$) and the expert's trajectories ($\tau_E$) through cross entropy loss. On the other hand, the policy $\pi_\theta$ is trained to fool the discriminator by generating trajectories that are close to the expert trajectories according to the discriminator. Since $\log D_\phi(s, a)$ is non-differentiable with respect to $\theta$ in Equation 5, the score function estimator is used to compute the gradient, which leads to a form of policy gradient (Equation 6) using the discriminator as a reward function.

It has been shown that GAIL amounts to minimizing the Jensen-Shannon divergence between the policy's occupancy measure and the expert's (Ho & Ermon, 2016; Goodfellow et al., 2014) as follows:

$$\underset{\theta}{\text{argmin}} \, \underset{\phi}{\text{argmax}} \, \mathcal{L}_{\text{GAIL}}(\theta, \phi) = \underset{\theta}{\text{argmin}} \, D_{\text{JS}}(\rho_{\pi_\theta} || \rho_{\pi_E}) - \lambda \mathcal{H}(\pi_\theta) \tag{7}$$

where $D_{\text{JS}}(p||q) = D_{\text{KL}}(p||(p + q)/2) + D_{\text{KL}}(q||(p + q)/2)$ denotes Jensen-Shannon divergence, a distance metric between two distributions, which is minimized when $p = q$.

## 4 GENERATIVE ADVERSARIAL SELF-IMITATION LEARNING

The main idea of Generative Adversarial Self-Imitation Learning (GASIL) is to update the policy to imitate past good trajectories using GAIL framework (see Section 3.2 for GAIL). We describe the details of GASIL in Section 4.1 and make a connection between GASIL and reward learning in Section 4.2, which leads to a combination of GASIL with policy gradient in Section 4.3.

## 4.1 ALGORITHM

The keay idea of GASIL is to treat good trajectories collected by the agent as trajectories that the agent should imitate as described in Algorithm 1. More specifically, GASIL performs the following two updates for each iteration.

---

**Algorithm 1** Generative Adversarial Self-Imitation Learning

---

Initialize policy parameter $\theta$
Initialize discriminator parameter $\phi$
Initialize good trajectory buffer $\mathcal{B} \leftarrow \emptyset$
**for** each iteration **do**
    Sample policy trajectories $\tau_\pi \sim \pi_\theta$
    Update good trajectory buffer $\mathcal{B}$ using $\tau_\pi$
    Sample good trajectories $\tau_E \sim \mathcal{B}$
    Update the discriminator parameter $\phi$ via gradient ascent with:

$$\nabla_\phi \mathcal{L}_{\text{GASIL}} = \mathbb{E}_{\tau_\pi} \left[ \nabla_\phi \log D_\phi(s,a) \right] + \mathbb{E}_{\tau_E} \left[ \nabla_\phi \log(1 - D_\phi(s,a)) \right] \tag{8}$$

    Update the policy parameter $\theta$ via gradient descent with:

$$\nabla_\theta \mathcal{L}_{\text{GASIL}} = \mathbb{E}_{\tau_\pi} \left[ \nabla_\theta \log \pi_\theta(a|s) Q(s,a) \right] - \lambda \nabla_\theta \mathcal{H}(\pi_\theta),$$
$$\text{where } Q(s,a) = \mathbb{E}_{\tau_\pi} \left[ \log D_\phi(s,a) | s_0 = s, a_0 = a \right] \tag{9}$$

**end for**

---

**Updating good trajectory buffer ($\mathcal{B}$)** GASIL maintains a *good trajectory buffer* $\mathcal{B} = \{\tau_i\}$ that contains a few trajectories ($\tau_i$) that obtained high rewards in the past. Each trajectory consists of a sequence of states and actions: $s_0, a_0, s_1, a_1, ..., s_T$. We define 'good trajectories' as any trajectories whose the discounted sum of rewards are higher than expected return of the current policy. Though there can be many different ways to obtain such trajectories, we propose to store top-K episodes according to the return $R = \sum_{t=0}^{\infty} \gamma^t r_t$.

**Updating discriminator ($D_\phi$) and policy ($\pi_\theta$)** The agent learns to imitate good trajectories contained in the good trajectory buffer $\mathcal{B}$ using generative adversarial imitation learning. More formally, the discriminator ($D_\phi(s,a)$) and the policy ($\pi_\theta(a|s)$) are updated via the following objective:

$$\underset{\theta}{\arg\min} \, \underset{\phi}{\arg\max} \, \mathcal{L}_{\text{GASIL}}(\theta, \phi) = \mathbb{E}_{\tau_\pi} \left[ \log D_\phi(s,a) \right] + \mathbb{E}_{\tau_E \sim \mathcal{B}} \left[ \log(1 - D_\phi(s,a)) \right] - \lambda \mathcal{H}(\pi_\theta) \tag{10}$$

where $\tau_\pi, \tau_E$ are sampled trajectories from the policy $\pi_\theta$ and the good trajectory buffer $\mathcal{B}$ respectively. Intuitively, the discriminator is trained to discriminate between good trajectories and the policy's trajectories, while the policy is trained to make it difficult for the discriminator to distinguish by imitating good trajectories.

## 4.2 CONNECTION TO REWARD LEARNING

The discriminator in GASIL can be interpreted as a reward function for which the policy optimizes because Equation 9 uses the score function estimator to maximize the reward given by $-\log D_\phi(s,a)$. In other words, the policy is updated to maximize the discounted sum of rewards given by the discriminator rather than the true reward from the environment. Since the discriminator is also *learned*, GASIL can be viewed as an instance of optimal reward learning algorithm (Sorg et al., 2010). A potential benefit of GASIL is that the optimal discriminator can provide intermediate rewards to the policy along good trajectories, even if the true reward from the environment is delayed. In such a scenario, GASIL can allow the agent to learn more easily compared to the true reward function. Indeed, as we will show in Section 5.4, GASIL performs significantly better than a state-of-the-art policy gradient baseline in a delayed reward setting.

## 4.3 COMBINING WITH POLICY GRADIENT

As the discriminator can be interpreted as a learned internal reward function, it can be easily combined with any RL algorithms. In this paper, we explore a combination of GASIL objective and policy gradient objective (Equation 2) as follows:

$$\nabla_\theta J_{\text{PG}} - \alpha \nabla_\theta \mathcal{L}_{\text{GASIL}} = \mathbb{E}_{\pi_\theta} \left[ \nabla_\theta \log \pi_\theta(a|s) \hat{A}_t^\alpha + \lambda \nabla_\theta \mathcal{H}(\pi_\theta) \right] \tag{11}$$

where $\hat{A}_t^\alpha$ is an advantage estimation using a modified reward function $r^\alpha(s,a) \triangleq r(s,a) - \alpha \log D_\phi(s,a)$. Intuitively, the discriminator is used to shape the reward function to encourage the policy to imitate good trajectories.

## 5 EXPERIMENTS

The experiments are designed to answer the following questions: (1) What is learned by GASIL? (2) Is GASIL better than behavior cloning approach? (3) Is GASIL competitive to policy gradient method?; (4) Is GASIL complementary to policy gradient method when combined together?

### 5.1 IMPLEMENTATION DETAILS

We implemented the following agents:

- PPO: Proximal policy optimization (PPO) baseline (Schulman et al., 2017).
- PPO+BC: PPO with additional behavior cloning to top-K trajectories.
- PPO+SIL: PPO with self-imitation learning (Oh et al., 2018).
- PPO+GASIL: Our method using both the discriminator and the true reward (Section 4.3).

The details of the network architectures and hyperparameters are described in the appendix. Our implementation is based on OpenAI's PPO and GAIL implementations (Dhariwal et al., 2017).

### 5.2 2D POINT MASS

To better understand how GASIL works, we implemented a simple 2D point mass environment with continuous actions that determine the velocity of the agent in a 2D space as illustrated in Figure 2. In this environment, the agent should collect as many blue/green objects as possible that give positive rewards (5 and 10 respectively) while avoiding distractor objects (orange) that give negative rewards (-5). There is also an actuation cost proportional to L2-norm of action which discourages large velocity.

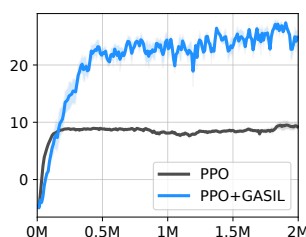

Figure 1: Learning curve on 2D point mass. See text for details.

The result in Figure 1 shows that PPO tends to learn a sub-optimal policy quickly. Although PPO+GASIL learns slowly in the early stage, it finds a better policy at the end of learning compared to PPO.

Figure 2 visualizes the learning progress of GASIL with the learned discriminator. It is shown that the initial top-K trajectories collect several positive objects as well as distractors on the top area of the environment. This encourages the policy to explore the top area because GASIL encourages the agent to imitate those top-K trajectories. As visualized in the third row in Figure 2, the discriminator learns to put higher rewards for state-actions that are close to top-k trajectories, which strongly encourages the policy to imitate such trajectories. As training goes and the policy improves, the agent finds better trajectories that avoid distractors while collecting positive rewards. The good trajectory buffer is updated accordingly as the agent collects such trajectories, which is used to train the discriminator. The interaction between the policy and the discriminator converges to a sub-optimal policy which collects two green objects.

In contrast, Figure 3 visualizes the learning progress of PPO. Even though the agent collected the same top-k trajectories at the beginning as in PPO+GASIL (compare the first columns of Figure 2 and Figure 3), the policy trained with PPO objective quickly converges to a sub-optimal policy which collects only one green object depending on initial positions. We conjecture that this is because the policy gradient objective (Eq 2) with the true reward function strongly encourages collecting nearby positive rewards and discourages collecting negative rewards. Thus, once the agent learns a sub-optimal behavior as shown in Figure 3, the true reward function discourages further exploration due to distractors (orange objects) nearby green objects and the actuation cost.

On the other hand, our GASIL objective does not explicitly encourage nor discourage the agent to collect positive/negative rewards, because the discriminator gives the agent internal rewards according to how close the agent's trajectories are to top-K trajectories regardless of whether it collects some objects or not. Though this can possibly slow down learning, it can often help finding a better policy in the end depending on tasks as shown in this experiment. This result also implies that directly learning to maximize true reward such as in the policy gradient method may not always lead to the best behavior due to the learning dynamics.

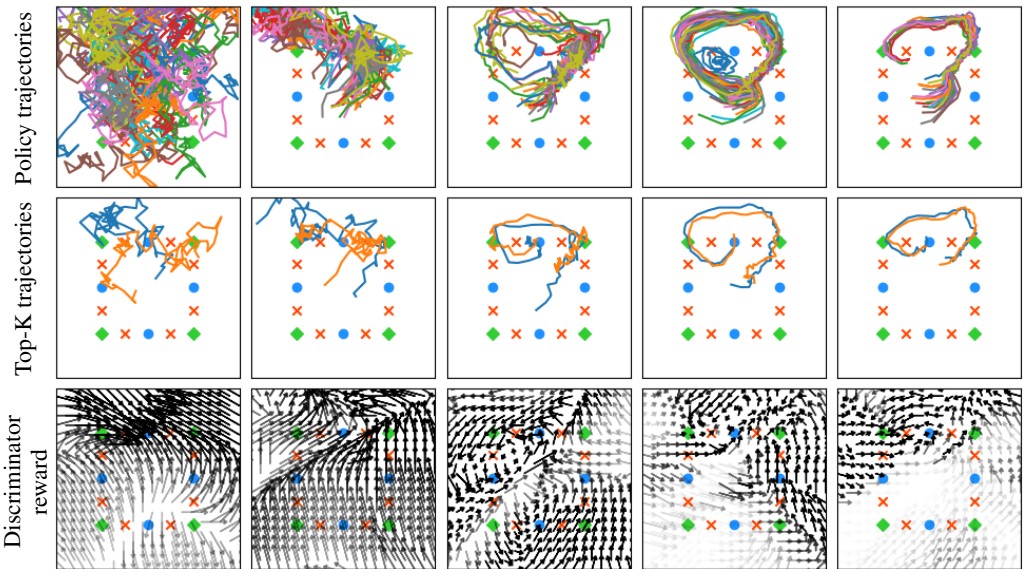

Figure 2: Visualization of GASIL policy on 2D point mass. The first two rows show the agent's trajectories and top-k trajectories at different training steps from left to right. The third row visualizes the learned discriminator at the corresponding training steps. Each arrow shows the best action at each position of the agent for which the discriminator gives the highest reward. The transparency of each arrow represents the magnitude of the discriminator reward (higher transparency represents lower reward).

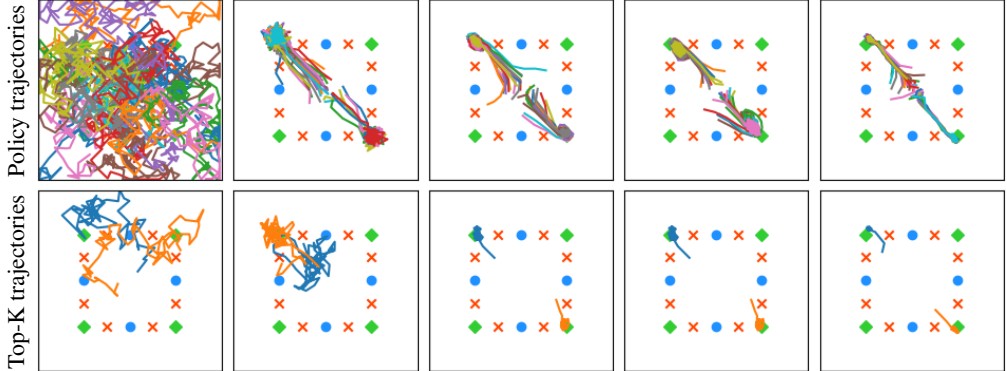

Figure 3: Visualization of PPO policy on 2D point mass. Compared to GASIL (Figure 2), PPO tends to prematurely learn a worse policy.

## 5.3 MuJoCo

To further investigate how well GASIL performs on complex control tasks, we evaluated it on OpenAI Gym MuJoCo tasks (Brockman et al., 2016; Todorov et al., 2012).[1] The result in Figure 4 shows that GASIL improves PPO on most of the tasks. This indicates that GASIL objective can be complementary to PPO objective, and the learned reward acts as a useful reward shaping that makes learning easier.

It is also shown that GASIL significantly outperforms the behavior cloning baseline ('PPO+BC') on most of the tasks. Behavior cloning has been shown to require a large amount of samples to imitate compared to GAIL as shown by Ho & Ermon (2016). This can be even more crucial in the RL context because there are not many good trajectories in the buffer (e.g., 1K-10K samples). Besides, GASIL also outperforms self-imitation learning ('PPO+SIL') (Oh et al., 2018) showing that our generative adversarial approach is more sample-efficient than self-imitation learning. In fact, self-imitation learning can be viewed as a type of behavior cloning with different sample weights according to their advantages, which can be the reason why GASIL is more sample-efficient. Another possible reason

---

[1] The demo video of the learned policies are available at `https://youtu.be/AwrtIUS2_pc`.

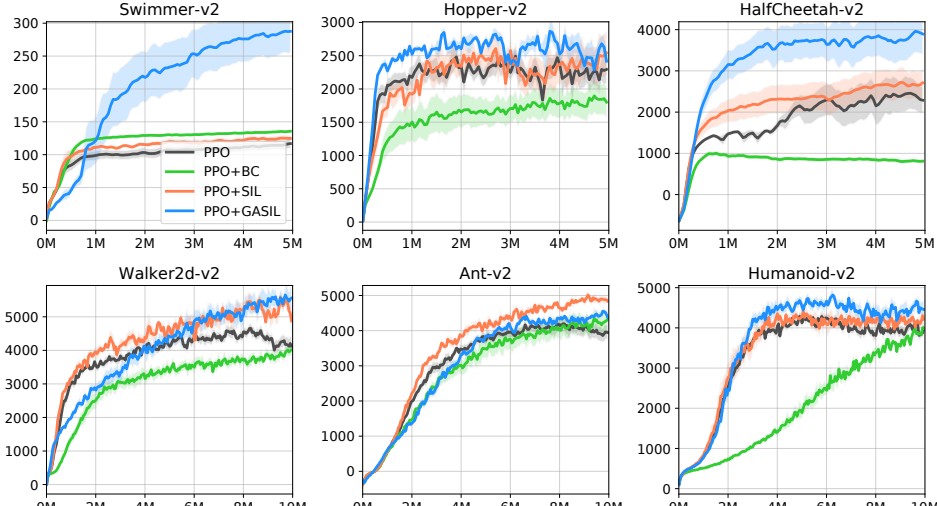

Figure 4: Learning curves on OpenAI Gym MuJoCo tasks averaged over 10 independent runs. x-axis and y-axis correspond to the number of steps and average reward.

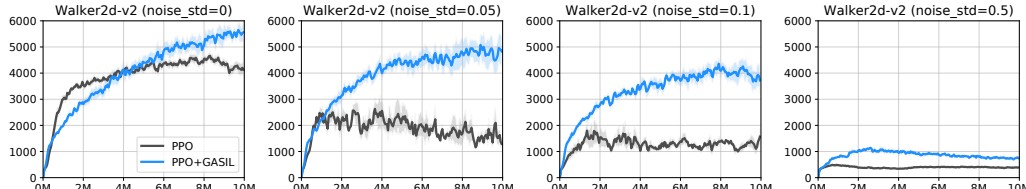

Figure 5: Learning curves on stochastic Walker2d-v2 averaged over 10 independent runs. The leftmost plot shows the learning curves on the original task without any noise in the environment. The other plots show learning curves on stochastic Walker2d-v2 task where Gaussian noise with standard deviation of $\{0.05, 0.1, 0.5\}$ (from left to right) is added to the observation for each step independently.

would be that GASIL generalizes better than behavior cloning method under non-stationary data (i.e., good trajectory buffer changes over time).

We further investigated how robust GASIL is to the stochasticity of the environment by adding a Gaussian noise to the observation for each step on Walker2d-v2. The result in Figure 5 shows that the gap between PPO and PPO+GASIL is larger when the noise is added to the environment. This result suggests that GASIL can be robust to stochastic environments to some extent in practice.

## 5.4 DELAYED MUJOCO

OpenAI Gym MuJoCo tasks provide dense reward signals to the agent according to the agent's progress along desired directions. In order to see how useful GASIL is under more challenging reward structures, we modified the tasks by delaying the reward of MuJoCo tasks for 20 steps. In other words, the agent receives an accumulated reward only after every 20 steps or when the episode terminates. This modification makes it much more difficult for the agent to learn due to the delayed reward signal.

The result in Figure 6 shows that GASIL is much more helpful on delayed-reward MuJoCo tasks compared to non-delayed ones in Figure 4, improving PPO on all tasks by a large margin. This result demonstrates that GASIL is useful especially for dealing with delayed reward because the discriminator gives dense reward signals to the policy, even though the true reward is extremely delayed.

## 5.5 EFFECT OF HYPERPARAMETERS

Figure 7 shows the effect of GASIL hyperparameters on Walker2d-v2. Specifically, Figure 7a shows the effect of the size of good trajectory buffer in terms of maximum steps in the buffer. It turns out that the agent performs poorly when the buffer size is too small (500 steps) or large (5000 steps).

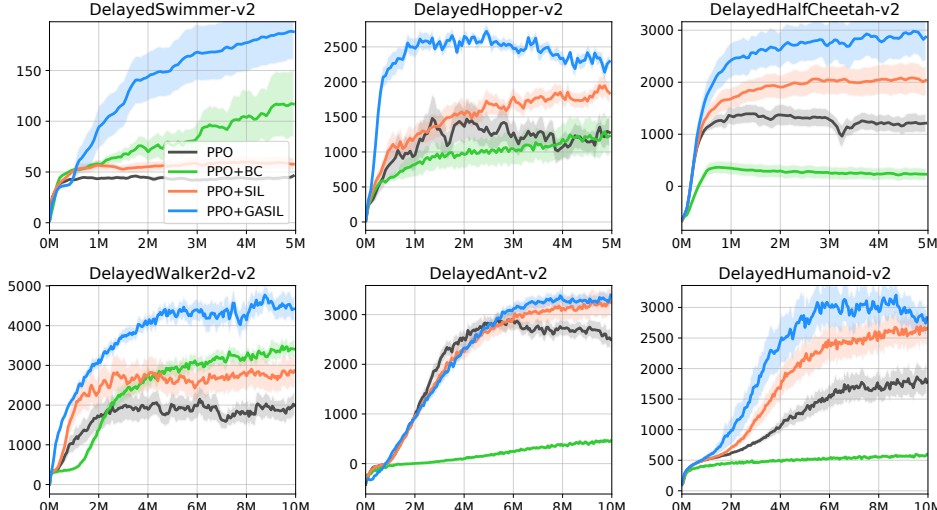

Figure 6: Learning curves on delayed-reward versions of OpenAI Gym MuJoCo tasks averaged over 10 independent runs. x-axis and y-axis correspond to the number of steps and average reward.

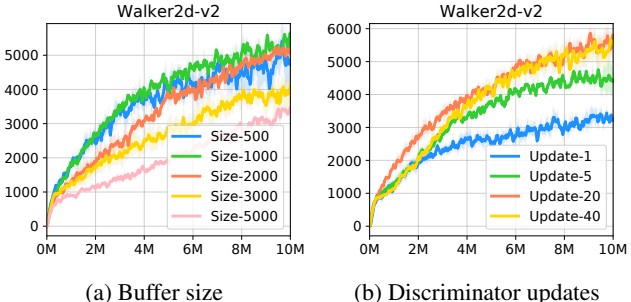

(a) Buffer size      (b) Discriminator updates

Figure 7: Effect of GASIL hyperparameters.

Although it is always useful to have more samples for imitation learning in general, the average return of good trajectories decreases as the size of the buffer increases. This indicates that there is a trade-off between the number of samples and the quality of good trajectories.

Figure 7b shows the effect of the number of discriminator updates with a fixed number of policy updates per batch. It is shown that too small or too large number of discriminator updates hurts the performance. This result is also consistent with GANs (Goodfellow et al., 2014), where the balance between the discriminator and the generator (i.e., policy) is crucial for the performance.

## 6 DISCUSSIONS

**Alternative ways of training the discriminator** We presented a simple way of training the discriminator: discriminating between top-K trajectories and policy trajectories. However, there can be many different ways of defining good trajectories and training the discriminator. Developing a more principled way of training the discriminator with strong theoretical guarantees would be an important future work.

**Dealing with multi-modal trajectories** In the experiment, we used a Gaussian policy with an independent covariance. This type of parameterization has been shown to have difficulties in learning diverse behaviors (Haarnoja et al., 2018; 2017). In GASIL, we observed that the good trajectory buffer ($\mathcal{B}$) often contain multi-modal trajectories because they are collected by different policies with different parameters over time. We observed that a Gaussian policy struggles with imitatating them reliably. In fact, there has been recent studies (Hausman et al., 2017; Li et al., 2017) that aim to imitate multi-modal behaviors using the GAIL framework. We believe that combining such methods would further improve the performance.

**Model-based approach**  We used a model-free GAIL framework which requires policy gradient for training the policy. However, our idea can be extended to model-based GAIL (MGAIL) (Baram et al., 2017) where the policy is updated by directly backpropagating through a learned discriminator and a learned dynamics model. Since MGAIL has been shown to be more sample-efficient than GAIL, we expect that a model-based counterpart of GASIL would also improve the performance.

## 7 CONCLUSION

This paper proposed Generative Adversarial Self-Imitation Learning (GASIL) as a simple regularizer for RL. The main idea is to imitate good trajectories that the agent has collected using generative adversarial learning framework. We demonstrated that GASIL significantly improves existing state-of-the-art baselines across many control tasks especially when rewards are delayed. Extending this work towards a more principled generative adversarial learning approach with theoretical guarantee would be an interesting research direction.

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

## A  EXPERIMENTS ON ATARI GAMES

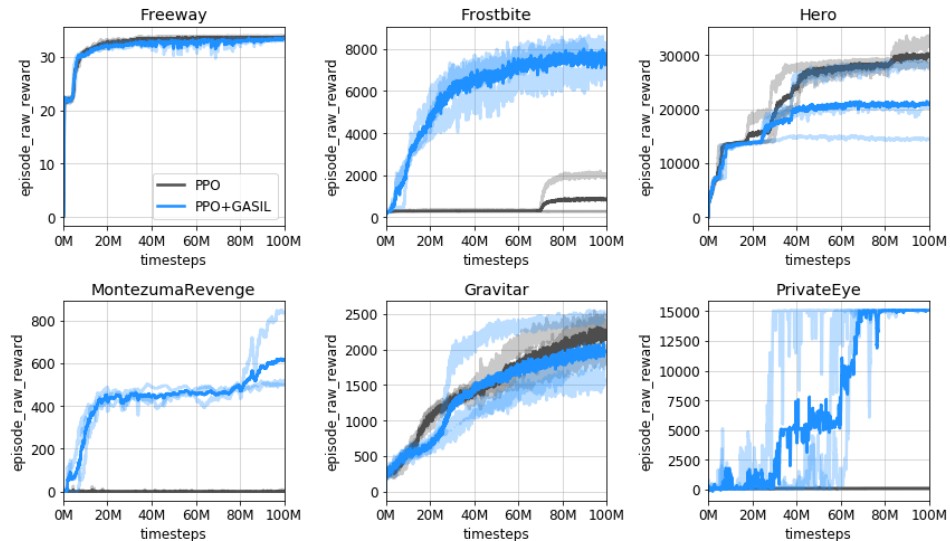

Figure 8: Learning curves on hard exploration Atari games averaged over 3 independent runs. x-axis and y-axis correspond to the number of steps and average reward.

|  | Montezuma | Freeway | Hero | PrivateEye | Gravitar | Frostbite |
|---|---|---|---|---|---|---|
| PPO | 20 | **34** | 30645 | 145 | 2406 | 915 |
| PPO+GASIL (Ours) | 629 | **34** | 21830 | **15099** | 2141 | **8276** |
| A2C+SIL (Oh et al., 2018) | **2500** | **34** | **33069** | 8684 | **2722** | 6439 |

Table 1: Compariston to A2C+SIL (Oh et al., 2018) on hard exploration Atari games.

To see how well GASIL performs with richer observation space, we evaluated it on hard exploration Atari games used in Oh et al. (2018). The result in Figure 8 shows that GASIL significantly improves PPO on Frostbite, Montezuma's Revenge, and PrivateEye. This suggests that GASIL is a useful RL regularizer that can be generally applied to a variety of domains. We further compared PPO+GASIL with A2C+SIL (Oh et al., 2018) in Table 1. It turns out that PPO+GASIL does not outperform A2C+SIL, though this is not a fair comparison as they use different actor-critic algorithms (i.e., A2C and PPO). In fact, GAIL (Ho & Ermon, 2016) has not been shown to be efficient on this type of domain where observations are high-dimensional. Thus, we conjecture that GASIL is more beneficial than SIL particularly for dealing with continuous control and simple observation space as shown in our MuJoCo experiments.

## B  HYPERPARAMETERS

Hyperperameters and architectures used for MuJoCo experiments are described in Table 2. We performed a random search over the range of hyperparameters specified in Table 2. For GASIL+PPO on Humanoid-v2, the policy is trained with PPO ($\alpha = 0$) for the first 2M steps, and $\alpha$ is increased to 0.02 until 3M steps. For the rest of tasks including all delayed-MuJoCo tasks, we used used a fixed $\alpha$ throughout training.

Table 2: GARL hyperparameters on MuJoCo.

| Hyperparameters | Value |
| --- | --- |
| Architecture | FC(64)-FC(64) |
| Learning rate | {0.0003, 0.0001, 0.00005, 0.00003} |
| Horizon | 2048 |
| Number of epochs | 10 |
| Minibatch size | 64 |
| Discount factor ($\gamma$) | 0.99 |
| GAE parameter | 0.95 |
| Entropy regularization coefficient ($\lambda$) | 0 |
| Discriminator minibatch size | 128 |
| Number of discriminator updates per batch | {1, 5, 10, 20} |
| Discriminator learning rate | {0.0003, 0.0001, 0.00002, 0.00001} |
| Size of good trajectory buffer (steps) | {1000, 10000} |
| Scale of discriminator reward ($\alpha$) | {0.02, 0.1, 0.2, 1} |

