# OpenReview forum: "Generative Adversarial Self-Imitation Learning"
_ICLR.cc/2019/Conference_

### Official Review · AnonReviewer1 · 2018-11-02
**combination of GAIL and self-imitation learning, but not convincing**

**Rating:** 5
**Confidence:** 4

**Review:**

This paper presents an incremental extension to the Self-imitation paper by Oh, Junhyuk, et al. The previous paper combined self-imitation learning with actor-critic methods, and this paper directly integrates the idea into the generative adversarial imitation learning framework.

I think the idea is interesting, but there remains some issues very unclear to me. In the algorithms, when updating the good trajectory buffer, it is said "We define ‘good trajectories’ as any trajectories whose the discounted sum of rewards are higher than that of the policy". What does "that of the policy" mean? How do you know the reward of the policy?

Second, without defining good trajectories, I don't think Algorithm 1 would work. Algorithm ` 1 misses the part of how to update buffer B. After introducing their own algorithm, the author did not provide much solid proof or analysis for why this self-imitation learning works.

In the experiment section, the author implemented GASIL for various applications and presented reasonable results and compared them with other methods. Nevertheless, without theoretical proof, it is hardly convincing that the results could be consistently reproduced instead of being merely accidental for some applications.

Update:
The rebuttal resolves some of my concerns. However, I still think the contribution is incremental. The current version looks too heuristic, more theoretical analysis or inspirations need to be added.

---

> ### Author Response · Authors · 2018-11-26
> **Response to R3**
>
> We thank Reviewer 3 for the detailed reviews and constructive feedback. We answer some questions specifically raised by the reviewer below. Please check the common response as well as our revised paper.
>
> - Regarding good trajectory buffer
> Thank you for pointing out unclear statements in the paper. Ideally, the good trajectories in the buffer are trajectories whose the discounted sum of rewards are greater than (or equal to) that of trajectories generated by the current policy in expectation. In practice, we proposed to store K-best trajectories (e.g., highest episode returns) in the buffer that the agent has collected during training. We observed that K-best trajectories satisfy the constraint described above with a proper K in all of our experiments. We have revised the paper to make this clear.

---

> > ### Comment · AnonReviewer1 · 2018-12-08
> > **thanks for the response**
> >
> > I mostly agree with the rebuttal. I have updated my score, but still think the contribution is not good enough to pass the high bar of ICLR. The paper would be stronger if more theoretical aspects are provided.

---

### Official Review · AnonReviewer2 · 2018-11-02
**Good natural algorithm but significance and understanding of when to apply is unclear.**

**Rating:** 6
**Confidence:** 5

**Review:**

Summary:

This paper proposes a self-imitation learning technique which modifies GAIL such that top-k trajectories with high reward found by the agent are kept in a buffer (and updated as learning goes on) such that the discriminator tries to distinguish between trajectories generated by the generator and those in the buffer while the generator tries to fool the discriminator by trying to imitate trajectories present in the buffer.

Experiments are shown on two domains: 1. a simple 2D domain where the agent must avoid orange circles (negative reward) and touch green and blue circles which yield positive reward and 2. on MuJoCo against PPO and variants of PPO as baselines where it is shown that GASIL performs better (even under increased stochastic noise in the dynamics.)


Comments:

- Generally well-written and easy to understand. Thanks!!

- Intuitive algorithm and good experiments with ablation studies on MuJoCo.

- My main concern is that the paper while offering a good self-imitation algorithm fails to really shine light on when/why this is expected to work. Especially in the following natural areas:

a. Why is it that performance decreases as buffer size B increases?
b. Why doesn't the policy get stuck imitating the first few good trajectories? the conjecture offered is that policy gradient strongly encourages greedy myopic behavior while GASIL does not. Wouldn't one expect GASIL to suffer more?
c. Does GASIL work better on rich observation spaces (e.g. Atari games) as well?

Without good answers (theoretical or empirical) to the above questions it is a bit hard to assess how significant of an improvement GASIL actually is and what is the prescription for using this over non-GAIL style self-imitation learning algorithms?

---

> ### Author Response · Authors · 2018-11-26
> **Response to R2**
>
> We thank Reviewer 2 for the detailed reviews and constructive feedback. We answer some questions specifically raised by the reviewer below. Please check the common response as well as our revised paper.
>
> - Regarding buffer size
> There is a trade-off between the number of samples in the buffer and the quality (i.e., average return) of the trajectories in the buffer, because the data in the buffer is collected by the agent as opposed to experts in standard imitation learning setup. More specifically, as the size of buffer increases, the average return of trajectories in the buffer generally decreases, while the samples become more diverse. So, the agent would not perform well if the buffer size is too small (due to lack of samples) or too large (due to low-quality data). To just clarify, the performance does not always decrease as the buffer size decreases (1000 is better than 500 in Figure 7a) in our experiment.

---

> > ### Comment · AnonReviewer2 · 2018-12-01
> > **Thanks for the clarification!**
> >
> > Thanks! I buy the explanation.

---

### Official Review · AnonReviewer3 · 2018-11-04
**No comparison/evaluation on discrete action tasks (i.e. ATARI games)**

**Rating:** 5
**Confidence:** 5

**Review:**

[Paper Summary]:
This paper proposes a regularization technique for existing RL algorithms by encouraging them to learn to reproduce the best past trajectories which obtained higher reward than that of current policy. The proposed method in the paper has the same high-level idea as "Self-imitation learning" [Oh et.al. ICML 2018] with a different objective. Instead of performing imitation learning to distill the knowledge from past best trajectories, this proposes to use inverse reinforcement learning via GAIL objective [Ho and Ermano, 2016]. The best k trajectories from past experience are stored to train a discriminator which is then used to augment the external reward function with a discriminator reward.

[Paper Strengths]:
The paper combines ideas from GAIL and self-imitation learning to propose a method that leverages past best trajectories via inverse-RL. This combination allows one to interpret self-imitation of best trajectories as a mechanism for "reward shaping" where learned discriminator shapes the environmental reward using past experiences. This is an exciting perspective and needs further discussion.

[Paper Weaknesses and Clarifications]:
=> This paper is very closely related to self-imitation learning [Oh et.al.], however, there is no theoretical justification provided (unlike [Oh et. al.]) whether the policy learned by optimizing Equation-11 is in anyway related to the optimal policy -- which was the case as shown in [Oh et.al.]. That being said, this is not a requirement for a paper to show theoretical justification as long as the paper justifies given approach with ample empirical evidence.
=> The main comparison point for the proposed approach, "GASIL", is "SIL" [Oh et.al.]. This paper provides a good comparison on continuous control tasks on Mujoco where "GASIL" performs slightly better than "SIL" in 3 out of 6 environments. However, "SIL" [Oh et. al.] showed extensive experiments on all Atari Games + Mujoco tasks. Since the proposed approach is mainly empirically motivated, the experiments should at least show a comparison on all the environments of the closely-related prior work. It would be much more convincing to see a bar chart across all 48 Atari Games showing relative improvement of "GASIL" over "SIL", as shown in Figure-4 of [Oh et. al.].
=> Other concerns:
    - The paper mentions on multiple occasions that the proposed method would handle delayed and "sparse" reward. However, it is not clear how can past best trajectories help with "sparse" rewards ("delayed-dense-rewards" seems alright, but they are not the same as "sparse"!). For instance, suppose the agent gets only terminal-reward in a maze. In such a case, the agent would need to rely on some form of exploration bonus (count-based, curiosity etc.) to reach the sparse-goal even once.
    - What prevents the learned policy from over-fitting to the local minima of the "locally" best trajectories seen so far?

[Final Recommendation]:
I request the authors to address the comments raised above. The paper has good potential, but sufficient empirical evidence is needed to justify the proposed technique. If the results on all Atari games can be included and shown to improve over "SIL", I would update my final rating.

---

> ### Author Response · Authors · 2018-11-26
> **Response to R1**
>
> We thank Reviewer 1 for the detailed reviews and constructive feedback. We answer some questions specifically raised by the reviewer below. Please check the common response as well as our revised paper.
>
>
> - Regarding exploration bonus and sparse reward
> Thank you for the great point. We removed the term “sparse” to prevent confusion. Our method does not improve the exploratory behavior of the agent. Thus, if the agent fails to reach the sparse goal even once, GASIL would not improve the performance because there is no good trajectory to imitate. For this type of extremely sparse-reward task, advanced exploration methods (e.g., count-based exploration, stein-variational-policy-gradient) would be necessary to discover reward signals, which is not addressed in our paper. However, once the agent reaches the goal, GASIL would exploit such rare experiences and encourage the agent to reproduce the same behavior much more easily compared to baseline algorithms (e.g., PPO). A similar discussion was made by the related work on self-imitation learning [Oh et al.].

---

> ### Comment · AnonReviewer3 · 2018-12-08
> **Response to author's rebuttal**
>
> The results provided on the ATARI games are not apples-to-apples with SIL[Oh et.al.], the baseline uses A2C and this paper uses PPO. Moreover, even in these comparisons, SIL[Oh et.al.] performs better on 4/6 games.
>
> Upon reviewing the author's responses and the update paper, I decided to keep my score the same. The paper may have good potential, but sufficient empirical evidence is needed to justify the proposed technique.

---

### Author Response · Authors · 2018-11-26
**Common response to all reviewers**

- Regarding the lack of result on Atari games
We have updated the paper (Appendix A) with results on 6 hard exploration Atari games as discussed in [Oh et al.]. We have observed that PPO+GASIL significantly improves PPO on 3 out of 6 hard Atari games. This shows that GASIL is a useful RL regularizer that can be applied to a variety of domains with rich observations. On the other hand, we observed that the overall result is not clearly better than A2C+SIL [Oh et al.], though this is not a fair comparison as the actor-critic algorithms are different (PPO, A2c). In fact, GAIL [Ho et al.] has not been shown to be efficient on this type of domain. Thus, we conjecture that GASIL is more beneficial than SIL particularly for continuous control as shown in our MuJoCo experiments.

- Regarding the lack of theoretical result
Our preliminary investigation shows that GASIL can guarantee policy improvement if 1) the average return of the trajectories in the buffer is higher than that of the agent’s policy, and 2) there exists a valid policy (i.e., occupancy measure) such that the trajectories in the buffer are unbiased samples from it. However, we did not manage to show that there exists a valid policy that can generate any trajectories in the buffer. Thus, we leave this for future work and present GASIL as an empirical study.

- Regarding overfitting to local minima and exploration
Not only GASIL but also any policy gradient algorithms generally do not guarantee convergence to the global optimal policy and rely on the stochasticity of the policy for exploration (e.g., e-greedy, dithering, entropy regularization). In practice, however, we observed that the buffer in GASIL tends to be constantly updated with better trajectories as the policy improves and uses a form of exploration (e.g., stochasticity encouraged by entropy regularization). Introducing a better form of exploration (e.g., curiosity-based exploration bonus, SVPG [Liu et al.]) would improve the performance of any policy gradient algorithms including ours. Thus, we believe that this is an orthogonal research direction.

---

### Meta-Review · Area_Chair1 · 2018-12-16
**combination of self-imitation and GAIL - needs more thorough development of conceptual insights**

**Confidence:** 4
**Recommendation:** Reject

**Metareview:**

The paper proposes an extension to reinforcement learning with self-imitation (SIL)[Oh et al. 2018]. It is based on the idea of leveraging previously encountered high-reward trajectories for reward shaping. This shaping is learned automatically using an adversarial setup, similar to GAIL [Ho & Ermon, 2016]. The paper clearly presents the proposed approach and relation to previous work. Empirical evaluation shows strong performance on a 2D point mass problem designed to examine the algorithms behavior. Of particular note are the insightful visualizations in Figure 2 and 3 which shed light on the algorithm's learning behavior. Empirical results on the Mujoco domain show that the proposed approach is particularly strong under delayed-reward (20 steps) and noisy-observation settings.

The reviewers and AC note the following potential weaknesses: The paper presents an empirical validation showing improvements over PPO, in particular in Mujoco tasks with delayed rewards and with noisy observations. However, given the close relation to SIL, a direct comparison with the latter algorithm seems more appropriate. Reviewers 2 and 3 pointed out that the empirical validation of SIL was more extensive, including results on a wide range of Atari games. The authors provided results on several hard-exploration Atari games in the rebuttal period, but the results of the comparison to SIL were inconclusive. Given that the main contribution of the paper is empirical, the reviewers and the AC consider the contribution incremental.

The reviewers noted that the proposed method was presented with little theoretical justification, which limits the contribution of the paper. During the rebuttal phase, the authors sketched a theoretical argument in their rebuttal, but noted that they are not able to provide a guarantee that trajectories in the replay buffer constitute an unbiased sample from the optimal policy, and that policy gradient methods in general are not guaranteed to converge to a globally optimal policy. The AC notes that conceptual insights can also be provided by motivating algorithmic or modeling choices, or through detailed analysis of the obtained results with the goal to further understanding of the observed behavior. Any such form of developing further insights would strengthen the contribution of the submission.